# Ex Vivo Fluorescence Confocal Microscopy (FCM) of Prostate Biopsies Rethought: Opportunities of Intraoperative Examinations of MRI-Guided Targeted Biopsies in Routine Diagnostics

**DOI:** 10.3390/diagnostics12051146

**Published:** 2022-05-05

**Authors:** Karl-Dietrich Sievert, Torsten Hansen, Barbara Titze, Birte Schulz, Ahmad Omran, Lukas Brockkötter, Alfons Gunnemann, Ulf Titze

**Affiliations:** 1Department of Urology, University Hospital OWL of the University of Bielefeld, Campus Lippe, 32756 Detmold, Germany; karl-dietrich.sievert@klinikum-lippe.de (K.-D.S.); ahmad.omran@klinikum-lippe.de (A.O.); lukas.brockkoetter@klinikum-lippe.de (L.B.); alfons.gunnemann@klinikum-lippe.de (A.G.); 2Institute of Pathology, University Hospital OWL of the University of Bielefeld, Campus Lippe, 32756 Detmold, Germany; torsten.hansen@klinikum-lippe.de (T.H.); barbara.titze@klinikum-lippe.de (B.T.); birte.schulz@klinikum-lippe.de (B.S.)

**Keywords:** confocal microscopy, prostate cancer, targeted biopsies, digital pathology

## Abstract

Background: The diagnosis of prostate carcinoma (PCa) requires time- and material-consuming histopathological examinations. Ex vivo fluorescence confocal microscopy (FCM) can detect carcinoma foci in diagnostic biopsies intraoperatively. Methods: MRI-guided and systematic biopsies were identified in a dataset of our previously published study cohort. Detection rates of clinically relevant tumors were determined in both groups. A retrospective blinded trial was performed to determine how many tumors requiring intervention were detectable via FCM analysis of MRI-guided targeted biopsies alone. Results: MRI-guided targeted biopsies revealed tumors more frequently than systematic biopsies. Carcinomas in need of intervention were reliably represented in the MRI-guided biopsies and were identified in intraoperative FCM microscopy. Combined with serum PSA levels and clinical presentation, 91% of the carcinomas in need of intervention were identified. Conclusions: Intraoperative FCM analysis of MRI-guided biopsies is a promising approach for the efficient diagnosis of PCa. The method allows a timely assessment of whether a tumor disease requiring intervention is present and can reduce the psychological stress of the patient in the waiting period of the histological finding. Furthermore, this technique can lead to reduction of the total number of biopsies needed for the diagnosis of PCa.

## 1. Introduction

PCa is the second most common cancer in men worldwide [1]. The clinical suspicion is based on the anamnesis, digital palpation examination and serum levels of prostate-specific antigen (PSA). The diagnosis is confirmed via histopathological examination of prostate biopsies. The conventional standard procedure (for initial biopsy) is the systematic collection of ten to twelve biopsies from defined areas of the prostate guided by transrectal ultrasound (systematic biopsies) [2]. All given information result in a therapy recommendation in the interdisciplinary tumor board. The risk stratification is based on histologic findings (lesion size and cancer grading according to the International Society of Urological Pathology (ISUP)), serum PSA and clinical and imaging findings [3]. Interventional therapies such as radical prostatectomy (RPE) or radiation are recommended for localized tumors in the intermediate- and high-risk groups [4]. Recent long-term outcomes showed that active surveillance (AS) is a safe option for localized low-risk tumors, aiming to reduce overtreatment and negative side effects of interventional treatments [5]. These patients follow up on a routine schedule with regular PSA measurements, digital rectal examinations and prostate biopsies to determine whether further AS is warranted or a progression of the tumor disease requiring interventional therapy is present.

Multiparametric magnetic resonance imaging (mpMRI) can be used to visualize differences in density and other structural changes of the prostate as well as areas with increased vascularization and blood flow. The procedure enables the detection and localization of carcinoma-susceptible focal findings [6]. The “Prostate Imaging Reporting and Data System” (PIRADS) is an internationally recognized standard used for the objective reporting of mpMRI data. Focal findings on mpMRI are categorized with respect to the likelihood of the presence of clinically significant prostate cancer using a 5-point scale (1–3 unlikely, 4 likely, 5 very likely) [7]. By overlaying the real-time ultrasound image with the MRI data, the suspect areas of the prostate can be targeted for biopsy (MRI-fused or -guided biopsies) [8]. Current guidelines recommend the combination of both procedures, especially in cases of a persistent suspected tumor and previous negative biopsy [4].

Quite a number of publications exist that evaluate the effects of PCa on mental health conditions. Side effects of treatments and the diagnosis itself may underlie the high prevalence of depressive and anxiety symptoms before, during and after treatment [9]. Body image [10], self-esteem [11] and sense of masculinity [12] are all impacted by a diagnosis of PCa. In addition, many men experience psychological problems during the course and time passing of related diagnostic procedures from initial PSA testing to the final treatment decisions [13]. These include so-called PSA-itis (anxiety associated with knowledge of the serum PSA level) [14] and PSA-dynia (a state of emotional or physical distress due to an elevated PSA level) [15]. Studies gave evidence that waiting for biopsy results was the most stressful event for 65% of men with PCa [16]. The Rotterdam PSA Screening Study found similar anxiety scores (STAI) among those patients who had undergone a biopsy procedure but had not yet received the results [17]. This fits with the finding that the highest serum cortisol levels were found in men who had undergone a prostate biopsy but were not informed of the result [18]. The NIH requires a histology to be submitted within 10 days after biopsy. However, this does not necessarily mean that the patient will be informed at that time. Typically, patients wait one to four weeks to receive prostate biopsy histology from their referring physician. Providing rapid (intraoperative) diagnostics after a TRUS-B at the site would be the most efficient way to reduce the psychological burden on patients.

Ex vivo confocal microscopy (FCM) is a novel imaging technique that provides optical digital sections of native biopsies and surgical resections [19,20]. The procedure is already used in dermatology in routine diagnostics for skin biopsies and excisions. Previous studies have shown that prostate carcinoma can be reliably detected and diagnosed with FCM in good agreement with final histology [21,22,23]. The specimens remain as unfixed tissue without loss and are available for subsequent histological, immunohistological and molecular examinations [24].

Based on the above considerations, the collected data were subjected to retrospective analysis to evaluate the impact of intraoperative FCM analysis of targeted biopsies from visible lesions in MRI. The first goal of this study was to discover the extent to which the FCM of targeted biopsies could predict the therapy recommendation of the tumor boards. The secondary goal was to evaluate the extent to which targeted and systematic biopsies contributed to the final therapy recommendation.

## 2. Materials and Methods

### 2.1. Analyzed Patient Cohort

This was a retrospective analysis of the results of a previously published study of intraoperative FCM examination of prostate biopsies [25]. The study cohort included data from 34 patients who underwent MRI-fused biopsy sampling for the diagnosis of PCa (532 total biopsies, 426 FCM images) The data set of one patient (P10) had to be excluded due to lack of documentation and incorrect MRI fusion. Twenty-eight patients underwent prostate biopsy because of clinical suspicion of PCa, and 6 patients underwent follow-up biopsies as part of AS for previously known PCa. The majority of the biopsies taken (438/544, 80%) were analyzed intraoperatively with a Vivascope 2500 G4 ex vivo confocal microscope. The biopsies were examined by a pathologist in the operating room in blinded conditions without knowledge of the collection site (targeted vs. systematic biopsy). In comparison with the final histological examination, prostate carcinomas could be detected in the biopsies with a sensitivity of 85% and a specificity of 100%. ISUP grades 3–5 were reliably diagnosed. Minor limitations were found in the differentiation of ISUP grade 2 from ISUP grade 1.

### 2.2. Data Set

For each of the 544 biopsies, the result of the final histological examination was available. Here, the presence of tumor lesions, their size (in mm and degree of infiltration related to the length of the biopsy in percent) and the ISUP grade were documented (Figure 1). In the review of the surgical protocols, each biopsy was assigned whether it was a targeted biopsy (central or peripheral) from a focus visible on MRI or a systematic biopsy. Furthermore, MRI findings were also compiled with the PIRADS group and PSA level at the time of biopsy procedure for each patient.

Biopsy examinations took place between November 2019 and June 2020. Follow-up data were obtained from all patients for the current study. Histologic findings were obtained for all patients who underwent prostatectomy and compared to biopsy results.

### 2.3. Analysis of Clinically Relevant Carcinoma Infiltrates in the Biopsies

The definition of clinically relevant PCa is not consistent in existing published studies. Based on autopsy studies, ISUP grade 1 tumors with foci sizes > 0.5 mL were defined as clinically significant for RPE preparations [26]. According to 3D reconstruction studies, this corresponds to a focal size of 6 mm in needle biopsies [27]. Consistent with a large validation study, tumors with ISUP grade 2 or focal sizes > 6 mm were defined as clinically relevant for this study [28]. Each tumor-bearing biopsy was assigned as to whether it was a clinically relevant lesion based on the above definition. The frequencies of clinically relevant tumor foci in the targeted and systematic biopsies were compared.

### 2.4. Definition of Carcinoma Requiring 1ntervention

Interventional therapy included RPE and radiation therapies. Active surveillance was not evaluated as intervention. In addition to histologic criteria, clinical data (age, general condition) and laboratory parameters (serum PSA level) played a role in the decision to treat patients with interventional therapy. Carcinomas requiring intervention were defined according to the guidelines via ISUP grade (ISUP grade > 1), degree of infiltration in the biopsies (>50%) and/or PSA level in serum (>10 ng/mL).

### 2.5. Study Design

The intraoperative FCM diagnoses of the targeted biopsies (ranging from 2 to 14 biopsies) were available from each patient. Of these, foci with the largest diameters or highest ISUP grades were defined as the resulting intraoperative FCM diagnosis. Age, PSA level and the resulting FCM diagnosis were summarized and blindly assessed by an experienced urologist. The patients were assigned to three cohorts: 0—no tumor; 1—tumor with unclear therapeutic relevance; 2—tumor requiring intervention. These simulated intraoperative assessments were compared with the final recommendations of the tumor boards (0—no tumor; 1—active surveillance; 2—interventional therapy (RPE, radiation)). Recommendations based on the intraoperative FCM analysis of targeted biopsies and the tumor boards were statistically compared.

### 2.6. Statistical Analysis

The simulated intraoperative judgments based on the FCM diagnoses of the targeted biopsies and the recommendations of the tumor boards were analyzed in an error matrix. For FCM diagnoses, categories 0 (no tumor) and 1 (tumor of unclear therapeutic consequence) were combined; in these cases, there would have been no indication for interventional therapy at the time of biopsy collection. Similarly, for tumor board therapy recommendations, categories 0 (no tumor) and 1 (active surveillance) were contrasted with category 2 (interventional therapy). Sensitivity, specificity and the positive/negative predictive value was calculated. The level of agreement was measured using Cohen’s Kappa [29] and interpreted according to the categories of Landis and Koch [30].

## 3. Results

### 3.1. Biopsy Sampling Sites

A total of 252/532 (47.4%) biopsies represented MRI-guided biopsies from the visible lesions on MRI (targeted biopsies). Of these, 143/252 (56.7%) biopsies were obtained from central sections of the visible lesions, and 109/252 (43.3%) biopsies were from peripheral sections of the MRI foci. A total of 280/532 (52.6%) biopsies were systematic biopsies taken from all other sections of the prostate. These data show a stronger weighting of targeted biopsies compared with the recommendations of the current guidelines.

### 3.2. MRI Findings

The cohort included one patient (P09) with a PIRADS 2 lesion. In this case, the biopsy was taken at the patient’s explicit request, knowing the family history with evidence of advanced prostate cancer in a direct relative. A PIRADS 3 lesion was present in 7/34 patients, and 11/21 patients presented with a PIRADS 4 lesion. A total of 15/34 presented with a PIRADS 5 lesion. Increasing frequencies and rates of carcinoma requiring intervention were seen with higher PIRADS grades (Figure 2). The patient with a PIRADS 2 lesion and serum PSA < 10 ng/mL surprisingly showed prostate carcinoma in need of intervention (ISUP grade 3) in the biopsies. 

### 3.3. Detection Rates of Clinically Relevant Tumor Infiltrates

A total of 173 MRI-fused targeted biopsies (7.2 ± 2.7) and 170 standard biopsies (7.4 ± 4.1) were available from the 21 tumor patients (Table 1). Tumor foci were detectable in 65/173 (38%) of the MRI-guided targeted biopsies. Of these, 47/65 (72%) biopsies were clinically relevant tumor foci. In contrast, 18/170 (11%) of the systematic biopsies had tumor infiltrates. Of these biopsies, 11/18 (61%) showed clinically relevant tumor foci. In summary, we found a higher cancer detection probability in the targeted biopsies at the same rate of clinically relevant tumor manifestations. 

Clinically relevant tumor foci were present in the targeted biopsies in 12/14 patients with carcinomas requiring intervention. In 6/14 (43%) patients, these were exclusively detectable in the targeted biopsies, and 8/14 (57%) patients showed relevant tumor foci in both targeted biopsies and standard biopsies. From the 2 remaining, 1 patient showed relevant carcinoma in the systematic biopsies.

### 3.4. Therapy Decisions Based on the Targeted Biopsies

In the retrospective blinded synopsis of the findings from the targeted biopsies and the serum PSA values (Table 2), 13/14 (92%) patients were correctly diagnosed with tumors needing intervention: 10/14 patients (71%) were identified via an ISUP grade > 1. A total of 2/14 (14%) patients (P01, P24) showed an ISUP grade 1 tumor > 6 mm in size in the targeted biopsies; P24 showed a concomitant PSA elevation to >10 ng/mL. In 1/14 patients (P28), there was a borderline constellation between interventional therapy and AS, but in this case, considering the young age, a simulated intraoperative decision would have been made in favor of surgical therapy. 

In only 1/14 (7%) patients (P29) clinical data, serum PSA and intraoperative FCM diagnosis showed the constellation for AS. The indication for interventional therapy was made only in the interdisciplinary tumor conference. Although in this case the criteria for active surveillance were still fulfilled, the indication for surgical therapy was made in view of the family history (brother died of prostate cancer) and at the patient’s request.

### 3.5. Therapy Decisions in Patients under Active Surveillance

A total of 2/6 (AS) patients with known prostate cancer showed tumor progressions, and indications for interventional therapies were established. P01 suggested an increase in tumor size in the targeted biopsies already correlating with PSA increase. P03 revealed a higher ISUP grade in the targeted biopsies. The remaining 4/6 patients remained in AS. Of these, one patient (P20) showed localized tumor formations in one of nine targeted biopsies. In one patient (P31), tumor formations were exclusively recorded in the systematic biopsies. In the remaining patients (P06, P34), tumor foci did not appear in either the targeted or systematic biopsies.

A total of 3 pretherapeutic patients (P05, P14, P17) were eligible for AS. One of these patients (P05) had a PSA elevation to 11.6 ng/mL, so that in this case surgical therapy was discussed intraoperatively. However, since no further tumor foci were detectable in the further systematic biopsies, the tumor board decided in favor of AS. One patient (P17) showed a micro-focal ISUP grade 1 tumor in the targeted biopsies only. The other patient had small tumor infiltrates in one of the systematic biopsies but not in the targeted biopsies.

### 3.6. Follow-Up Results

Follow-up data were available of all patients who were classified to interventional therapy. Of these, 2/14 patients underwent radiotherapy (P03, P26), 1/14 patients (P12) had a metastasized course and underwent chemotherapy with androgen suppression and 11/14 patients underwent radical prostatectomy.

The histological evaluations of these prostatectomy specimens revealed ISUP grade 3 tumors in 4/11 cases, ISUP grade 2 tumors in 6/11 cases and one case with an ISUP grade 1 tumor. A total of 3/11 cases showed extra-prostatic invasion (2x pT3a, 1x pT3b) and lymph node metastasis in one case. P29 remained an unusual case showing only localized carcinoma in the prostatectomy specimen (pT2a, pNx, R0) but bone metastasis in the later course.

### 3.7. Summary

FCM analysis of targeted biopsies identified 13/14 PCa requiring intervention consistent with the tumor board (Table 3). One patient met criteria for AS but opted for RPE based on family history. No patient requiring intervention was identified solely on the basis of the systematic biopsies, and the remaining patients were either tumor-free or met criteria for AS.

## 4. Discussion

Ex vivo FCM examinations of prostate biopsies are a promising option for intraoperative detection of PCa in need of intervention. Compared with conventional histology, the examinations are less labor intensive and can be performed by one person in the operating room. However, the examination of all biopsies taken would be too time consuming in the intraoperative setting. For routine diagnostics, the biopsies examined should be carefully selected.

Data are available of the diagnostic accuracy of MRI-guided and systematic biopsies from systematic reviews and diagnostic studies [31,32,33]. Reported results and differences between these methods vary depending on study design and patient population. It should be noted that the definition of clinically significant carcinomas is not entirely uniform. Overall, MRI-assisted biopsies were shown to detect more significant PCa compared with conventional systematic biopsies. In a patient population with prostatectomy as the reference standard, the combination of both procedures was shown to be least likely to result in subsequent tumor upgrading to ISUP grade 3 or higher [34]. These findings were also confirmed in our cohort. With a higher PIRADS category, the frequency of detected carcinomas and the proportion of tumors requiring intervention increased. MRI-guided biopsies from visible foci were 3.5 times more likely to hit relevant tumor foci.

In consideration of this finding, the possibility for intraoperative examinations of targeted biopsies from the prostate is a very interesting prospect. To our knowledge, this retrospective study represents the first analysis of intraoperative FCM diagnoses of MRI-guided targeted biopsies. The rapid preliminary FCM diagnosis in conjunction with clinical parameters (patient age) and serum PSA levels identified 13/14 (sensitivity 93%, specificity 95%, Κ = 0.88) patients with a tumor disease requiring intervention. 

This approach allowed prompt preliminary grouping into patients with tumors requiring intervention and a second group of patients with lesions questionably requiring intervention or lacking a tumor. FCM examinations should be performed intraoperatively as an intermediate step without interfering with the final histologic and immunohistologic investigation. The final diagnosis was available after obtaining a conventional histology of both targeted and systematic biopsies. In our cohort, the systematic biopsies identified primarily AS patients, whereas no tumor foci requiring therapy were detected in any patient beyond the extent of the targeted biopsies. In AS patients with previously known PCa, tumor disease requiring intervention was reliably detectable via synopsis of the MRI-targeted biopsies with the levels of PSA, whereas a negative finding in the FCM examinations indicated that the patients could remain in AS.

The limitations of this study are the small sample size and that the MRI-fused biopsies were overrepresented in the present data set with 50% of all biopsies taken. Detection of fused glands in ISUP grade 2 tumors remains challenging, which leads to a high rate of discrepancies in second opinion investigations [35,36]. Therefore, a higher rate of upgrading in the final histology is to be expected if fewer biopsies are examined intraoperatively than in the present study. Furthermore, this is a retrospective evaluation, and the results should be verified in a prospective approach with a bigger cohort. 

In summary, intraoperative FCM analysis of MRI-fused targeted biopsies from suspicious foci of the prostate represents a promising approach for a more efficient diagnosis of prostate cancer. If a carcinoma requiring therapy is already diagnosed intraoperatively with a high degree of certainty (ISUP grades 3–5), this could eliminate the need for additional systematic biopsies [37]. The rate of septic complications [38] could be reduced, and local inflammation would also be minimized, which would be beneficial for the outcomes of the upcoming surgical procedures. 

From a clinical perspective, patients can be informed about upcoming interventional therapy while still in the hospital. The clinical colleagues can provide the desperately waiting patient with the necessary information in a timely manner in order to prepare promptly with his relatives for any further therapeutic and diagnostic steps. 

## Figures and Tables

**Figure 1 diagnostics-12-01146-f001:**
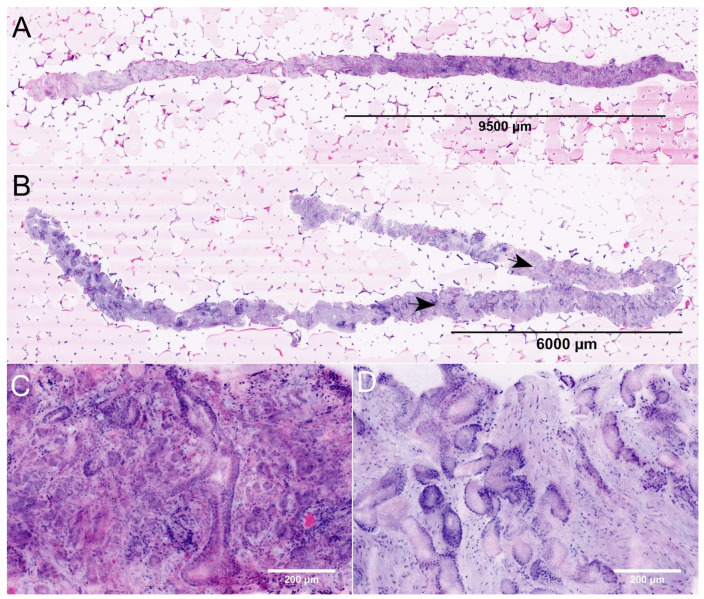
FCM images of MRI-guided biopsies of the prostate. (**A**) + (**C**): Full Scan (**A**) and detail (**C**) of an MRI-targeted biopsy from a PI-RADS-5 lesion (P22). The images show the biopsy of 19.8 mm length carrying carcinoma (lesion size 9.6 mm, infiltration grade 48%). The tumor focally shows patterns of poorly formed and fused glands (GLEASON-pattern 4). The intraoperative diagnosis was ISUP grade 2 carcinoma indicating a tumor disease requiring intervention. (**B**) + (**D**): Full scan (**B**) and detail (**D**) of an MRI-targeted biopsy from another PI-RADS-5 lesion (P31). This biopsy of 27.4 mm length contains a tumor infiltrate of 9.3 mm size (arrows, infiltration grade 34%). The tumor was consistently composed of well differentiated glands (GLEASON-pattern 3). The intraoperative diagnosis was ISUP grade 1 carcinoma. To date, the patient remains in active surveillance.

**Figure 2 diagnostics-12-01146-f002:**
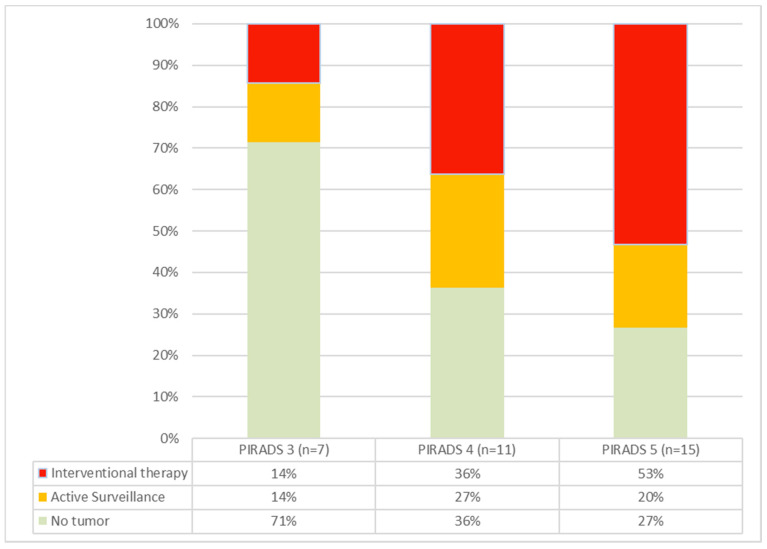
Cancer rates in PIRADS groups 3–5. As the PIRADS group ascends, both the incidence of detected carcinomas and the proportion of tumors requiring intervention increase.

**Table 1 diagnostics-12-01146-t001:** Representation of clinically relevant carcinoma in MRI-guided targeted biopsies and system biopsies.

Patient Data				Targeted Biopsies			Standard Biopsies		
Patient	Indication	PIRADS	Biopsies	Total	Tumor	Relevant	Total	Tumor	Relevant
P01	AS	4	24	8	4	1	16	3	0
P02	PRE	4	35	-	-	-	-	-	-
P03	AS	3	24	14	5	3	10	2	2
P04	PRE	5	30	-	-	-	-	-	-
P05	PRE	5	12	2	1	0	10	0	0
P06	AS	3	12	3	0	0	9	0	0
P07	PRE	5	12	6	4	4	6	1	0
P08	PRE	5	12	-	-	-	-	-	-
P09	PRE	2	12	6	5	5	6	0	0
P10	-	-	-	-	-	-	-	-	-
P11	PRE	4	12	7	3	2	5	0	0
P12	PRE	5	14	11	6	6	3	1	1
P13	PRE	3	12	7	-	-	-	-	-
P14	PRE	4	12	8	0	0	4	1	0
P15	PRE	3	12	7	-	-	5	-	-
P16	PRE	4	20	-	-	-	-	-	-
P17	PRE	4	21	3	1	0	18	0	0
P18	PRE	3	14	-	-	-	-	-	-
P19	PRE	3	14	7	0	0	7	0	0
P20	AS	5	12	9	1	1	3	0	0
P21	PRE	3	15	-	-	-	-	-	-
P22	PRE	5	13	8	6	6	5	3	2
P23	PRE	5	12	8	5	5	4	0	0
P24	PRE	4	12	7	3	1	5	2	1
P25	PRE	5	14	4	3	1	10	0	0
P26	PRE	5	14	9	9	9	5	3	3
P27	PRE	5	20	6	3	3	14	0	0
P28	PRE	4	16	6	5	0	10	1	1
P29	PRE	5	14	10	1	0	4	0	0
P30	PRE	5	15	-	-	-	-	-	-
P31	AS	5	12	7	0	0	5	1	1
P32	PRE	4	13	-	-	-	-	-	-
P33	PRE	4	15	-	-	-	-	-	-
P34	AS	4	16	10	0	0	6	0	0
P35	PRE	5	15	-	-	-	-	-	-
			Ʃ	173	65	47	170	18	11
			%		38%	72%		11%	61%

Indication for biopsy (PRE—pretherapeutic; AS—active surveillance known PCa), PIRADS group, number of biopsies in total as well as the numbers of targeted and systematic biopsies are shown for each patient. The absolute counts for detected carcinomas and clinically relevant tumor foci are presented. The rates of detected carcinomas were 3.5 times higher in the targeted biopsies. The proportion of clinically relevant carcinomas was comparable in both groups.

**Table 2 diagnostics-12-01146-t002:** Comparison of therapy recommendations based on the intraoperative FCM diagnoses and the tumor board combined with follow up data.

Clinical Data			Resulting FCM-Diagnosis of MRI Targeted Biopsies			Tumor Board					Follow Up		
Patient	Age	PSA	Length [mm]	ISUP Grade	Interv. (KDS)	Biopsies	Tumor	%	ISUP Grade	Intervention	Therapy	Stage	ISUP Grade
P01 ^+^	65	6.5	8	1	2	24	7	70	1	2	RPE	pT2c pN0 R0	2
P02	68	7.8	-	-	0	35	0	-	-	0	-	-	-
P03 ^+^	77	3.2	4	4	2	24	7	30	3	2	RT	n.a.	n.a.
P04	66	16	-	-	0	30	0	-	-	0	-	-	-
P05	79	11.6	4	1	2	12	1	25	1	1	AS	pT1c	1
P06 ^+^	60	8	-	-	0	12	0	-	-	1	AS	pT1c	1
P07	57	9	8.5	2	2	12	5	65	2	2	RPE	pT2c pN0 R0	3
P08	49	1.1	-	-	0	12	0	-	-	0	-	-	-
P09	57	6.4	10	4	2	12	5	90	3	2	RPE	pT3a pN0 R0	3
P10	-	-	-	-	-	-		-	-	-	-	-	-
P11	59	9.6	9.5	2	2	12	3	65	2	2	RPE	pT3a pNx R0	3
P12	79	55	14.5	5	2	14	7	90	5	2	CT	n.a. *	n.a. *
P13	64	11.8	-	-	0	12	0	-	-	0	-	-	-
P14	64	6.6	-	-	0	12	1	5	1	1	AS	pT1c	1
P15	61	3.55	-	-	0	12	0	-	-	0	-	-	-
P16	66	18.6	-	-	0	20	0	-	-	0	-	-	-
P17	78	4.68	0.1	1	1	21	1	10	1	1	AS	pT1c	1
P18	72	16.4	-	-	0	14	0	-	-	0	-	-	-
P19	74	14.55	-	-	0	14	0	-	-	0	-	-	-
P20 ^+^	58	5.96	8	1	1	12	1	45	1	1	AS	pT1c	1
P21	66	5.4	-	-	0	15	0	-	-	0	-	-	-
P22	66	9.92	17	2	2	13	9	95	2	2	RPE	pT3b pN1 R0	3
P23	73	8.7	13	4	2	12	5	70	4	2	RPE	pT2c pN0 R0	3
P24	69	12.9	6.5	1	2	12	5	40	1	2	RPE	pT2c pN0 R0	2
P25	77	18	8.8	2	2	14	3	40	2	2	RPE	pT2c pN0 R0	2
P26	61	33	14	2	2	14	12	55	2	2	RT	n.a.	n.a.
P27	64	32.7	7	2	2	20	3	95	2	2	RPE	pT2c pN0 R0	2
P28	52	6.39	3	1	2	16	6	45	1	2	RPE	pT2c pN0 R0	2
P29	63	5.89	2	1	1	14	1	10	1	2	RPE	pT2a pNx R0 *	1
P30	61	3.31	-	-	0	15	0	-	-	0	-	-	-
P31 ^+^	64	7.28	-	-	0	12	1	40	1	1	AS	pT1c	1
P32	63	4.41	-	-	0	13	0	-	-	0	-	-	-
P33	69	6.86	-	-	0	15	0	-	-	0	-	-	-
P34 ^+^	68	7.83	-	-	0	16	0	-	-	1	AS	pT1c	1
P35	55	3.81	-	-	0	15	0	-	-	0	-	-	-

Age, serum PSA, intraoperative FCM diagnoses of targeted biopsies are summarized for each patient. The derived judgements (0—no therapy, 1—no intervention/active surveillance, 2—interventional therapy) are contrasted with the final therapy recommendation of the tumor board. Almost all patients recommended for interventional therapy by the tumor board were identifiable via intraoperative FCM analysis of the targeted biopsies in combination with clinical data and serum PSA. Follow up data include the mode of therapy (RPE—radical prostatectomy, RT—radiation, CT—chemotherapy), tumor stage and final ISUP grade. ^+^ Patients with previously known PCa under active surveillance; * further follow up revealed metastasized clinical courses in these patients.

**Table 3 diagnostics-12-01146-t003:** Comparison of intraoperative estimations based on FCM analyses with final tumor board recommendations.

		FCM Targeted-Biopsies		
		No Tumor	Uncertain Relevance	Intervention
Tumorboard	No Tumor	*13*	*0*	*0*
		19		1
	Active Surveillance			
		*4*	*2*	*1*
	Intervention	*0*	*1*	13
		1		
Sensitivity		93%		
Specificity		95%		
Positive predictive value		93%		
Negative predictive value		95%		
Cohen’s Kappa		0.88		
Level of agreement		very good		

For FCM, categories 0 (no tumor) and 1 (tumor of unclear therapeutic consequence) were combined because there would have been no indication for interventional therapy (category 2) at the time of biopsy collection. Similarly, for tumor board therapy recommendations, categories 0 (no tumor) and 1 (active surveillance) were contrasted with category 2 (interventional therapy).

## Data Availability

Not applicable.

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
