# Peer review of "Ex Vivo Fluorescence Confocal Microscopy (FCM) of Prostate Biopsies Rethought: Opportunities of Intraoperative Examinations of MRI-Guided Targeted Biopsies in Routine Diagnostics"

_diagnostics, 2022, doi:10.3390/diagnostics12051146_

Round 1

Reviewer 1 Report

Authors present an interesting study on ex vivo Fluorescence Confocal Microscopy (FCM) of prostate biopsies with the implementation of intraoperative examinations of MRI-guided target biopsies in routine diagnostics. The authors need to be congratulated on the study, it covers a very relevant topic. Introduction is reasonable, stats and concomitant results are sound. The discussion is coherent and interprets the findings with current literature. In light of the small sample size, limitations are thorough and take this into account. Still, I would like to ask a few questions, which may further strengthen the discussion and add to the clinical value of FCM. In the discussion you state that this may be a promising option for intraoperative detection of PCa in need of intervention. In light of your findings, this assumption may be justified. However, and only recently, a study could show the benefits of active surveillance (AS) for eligible patients (PMID: 35053530). In light of these results, could you discuss how you see the clinical value of ex vivo FCM for patients under AS? As you further state that in your “cohort, the system biopsies identified primarily AS patients, whereas no tumor foci requiring therapy were detected in any patient beyond the extent of the target biopsies”, what patients do you think would benefit most from ex vivo FCM? As you can see Hagmann, and colleagues also used MRI targeted biopsies for follow-up. Where do you see the future of MRI-guided prostate biopsy? Is the implementation feasible in clinical routine?

Reviewer 2 Report

Authors should be congratulated for the great contribute to the challenging topic. All future prospective should lead to improve prostate cancer detection reducing investigations number and to create new and better algorithms to properly manage early stage PCa patients, avoiding overdiagnosis and overtreatment. Despite the interesting topic, several points warrant a mention:

  1. Which kind of interventional therapies were adopted? If radical prostatectomy (RP) was performed, did the histological examination on RP specimen confirmed the results of FCm technique?
  2. Authors should perform cost-benefits analysis to evaluate how the cost of this novel technique impact on active-surveillance costs.
  3. Authors should properly discuss the consequences of PCA diagnosis on patients’ mental health. PSA-dynia does not represent the only consequences of PCA diagnosis. Authors could read this novel paper on consequences of PCA on mental balance of PCA patients (DOI: 3390/ijerph19084825) .

Round 2

Reviewer 2 Report

Authors should be congratulated for the great contribute to the challenging topic. All future prospective should lead to improve prostate cancer detection reducing investigations number and to create new and better algorithms to properly manage early stage PCa patients, avoiding overdiagnosis and overtreatment. 
Authors improved the quality of the manuscript and they have fully answered our doubts. The paper is suitable for publication.